# Non-Integrin Laminin Receptors: Shedding New Light and Clarity on Their Involvement in Human Diseases

**DOI:** 10.3390/ijms26083546

**Published:** 2025-04-10

**Authors:** Filomena Napolitano, Maria Fabozzi, Nunzia Montuori

**Affiliations:** 1Department of Translational Medical Sciences, University of Naples Federico II, 80135 Naples, Italy; filomena.napolitano@unina.it (F.N.); maria.fabozzi@unina.it (M.F.); 2Center for Basic and Clinical Immunology Research (CISI), University of Naples Federico II, 80135 Naples, Italy

**Keywords:** laminin, non-integrin receptors, extracellular matrix, basement membrane

## Abstract

The extracellular matrix (ECM) is a dynamic network of macromolecules that provides structural support for cells and orchestrates cell signaling, functions, and morphology. The basement membrane constitutes a peculiar sheet-like type of ECM located between epithelial tissues and underlying connective tissues. The major constituent of the basement membrane is laminin, which exerts a remarkable repertoire of biological functions such as cell differentiation, migration, adhesion, and wound healing. Laminin performs its functions by interacting with two main classes of receptors, the integrin and the non-integrin laminin receptors, creating a complex network essential for tissue integrity and regeneration. Dysfunctional actions of laminin are the cause of diverse human diseases, including cancer, infectious, and neurodegenerative diseases. This topic has attracted researchers for some time, but the diversity of cell-surface receptors, through which laminin signaling occurs, makes the role of laminin controversial. Moreover, different laminin isoforms were identified, and each specific tissue basement membrane differs from the others in their laminin composition. This review focuses on the structural and molecular basis and pathophysiological relevance of specific interactions between laminins and non-integrin receptors in development, health, and disease.

## 1. Introduction

The structural support for cells and tissues is provided by the extracellular matrix (ECM), which is a dynamic three-dimensional network of different biological macromolecules including collagens, fibronectin, elastin, laminins, proteoglycans/glycosaminoglycans, and several other glycoproteins [1]. Based on the extracellular position, ECM is divided into two types: (i) the pericellular matrix, which creates an adhesive microenvironment for the resident cells, thus favoring cell attachment, and (ii) the interstitial matrix of connective tissue, which provides tissue integrity and support [1]. The basement membrane (BM) is a typical pericellular matrix with a sheet-like matrix surrounding most tissues and organs. BM is composed of laminin, type IV collagen, nidogen, and heparin sulfate proteoglycans, but only laminins are the first BM components appearing during the early phases of embryonic development [2]. Each laminin is a heterotrimer comprised of one of five α chains (encoded by LAMA1-5), one of four β chains (LAMB1-4), and one of three γ chains (LAMC1-3) joined together through a long coiled-coil domain to form a T- or cruciform-shaped structure [3]. The molecular weight of each laminin ranges from 400 to 900 kDa [3]. At the N-terminus, each laminin subunit contains various globular domains and epidermal growth factor-like (LE) domains, followed by a coiled-coil domain. Specifically, the globular domains comprise the N-terminal globular domain (LN), laminin 4 (L4) domains, and laminin four (LF) domains. α subunits are characterized by five laminin globular (LG) domains in the C-terminal end [4]. Compared to α subunits, β and γ subunits exhibit similar domains in the N-terminus but lack LG domains in the C-terminal [4]. Not all αβγ combinations are permissible, and only 16 different mammalian laminin isoforms have been detected. Laminin is named according to its subunit composition; for example, laminin-111 is composed of the subunits α1, β1, and γ1, while laminin-411 is composed of the subunits α4, β1, and γ1 [5]. α1, α5, β1, and γ1 are crucial for BM assembly during early embryogenesis, whereas γ1 subunit is common to most laminins [6]. The domain organization of laminin-111, used as an example of laminin structure, is reported in Figure 1.

Laminins represent an example of multifunctional molecules able to interact contemporarily both with other ECM macromolecules and multiple cell receptors by common structural motifs. On the one hand, laminin crosstalk with other components of BM by the N-terminus, thus creating a complex network for the architecture and physiology of BM. On the other hand, the C-terminal domains serve as a binding to specific receptors expressed by the plasma membranes of cells or microorganisms [4]. Laminin receptors are usually categorized into two types of surface receptors: integrins and non-integrin receptors. Upon laminin binding, integrins activate diverse signaling pathways to regulate cell growth, survival, migration, and differentiation. Integrins that have been identified as molecular partners for laminin include α1β1, α2β1, α3β1, α6β1, α10β1, α6β4, α7β1, and αvβ3 [7]. Little is known about the nature and functions of non-integrin receptors.

Through all these interactions, laminin exerts many important biological functions ranging from cell adhesion, shape, and movement to the promotion of tissue survival. Loss-of-function studies enriched our knowledge of laminin biology/function, especially in embryogenesis, vascular maturation, and neuromuscular development [8]. It is important to underline that laminin functions are dependent on specific laminin isoforms, but the molecular mechanism regulating laminin chain expression and the functions of each laminin subunit in different tissues have not yet been elucidated.

Given that laminin plays a central role in the assembly and physiology of BM, molecular defects caused by mutations, up- or downregulation, and alteration in its receptors can lead to severe pathological conditions. In fact, mutations of specific laminin chains cause muscular dystrophy, lethal skin blistering disease indicated as junctional epidermolysis bullosa, defects of the kidney filter and eye abnormalities (Pierson syndrome), and kidney, craniofacial, and limb development syndrome [9]. In addition to inherited diseases, cancer cells must pass through BM to reach the vascular system, and various pathogens enter the cells through direct interaction with laminin [10]. Thus, laminin is also implicated in cancer progression and in microbial and viral diseases. A recent comprehensive review on the role of laminin in cancer pathobiology has summarized the effects of laminins and their subunits in human cancers; alterations in the expression (up- or downregulation) or localization of specific laminins have been associated with the progression of several human cancers [11].

The laminin pattern via binding to integrins as well as the structural mechanism of laminin recognition by integrin has been well characterized [12]. In this review, we focus on the main non-integrin receptors and summarize the most recent knowledge on laminin/non-integrin receptor interaction across functional, structural, and pathological aspects.

## 2. Non-Integrin Laminin Receptors

In the last decades, diverse non-integrin receptors for laminin have been discovered, such as dystroglycan, 37/67 kDa laminin receptor, syndecan, Lutheran/basal cell adhesion molecule (Lu/BCAM), and melanoma cell adhesion molecule (MCAM). In general, the binding of these receptors to laminin occurs via LG4-5 domains of α subunits whereas LG1-3 domain interacts predominantly with integrins [4].

Structural, biochemical, and genetic analyses of these receptors have contributed to the elucidation of additional roles of laminin in various organ systems, beyond binding to integrin adhesion molecules. However, the functional significance of laminin/non-integrin receptor interaction is not yet fully known.

In the next sections, we describe the contribution of each non-integrin receptor to laminin functions, underlying their clinical significance in multiple human diseases, including infectious diseases, immune disorders, neurodegenerative diseases, and cancer. Future work investigating the non-integrin interactome of laminins will undoubtedly open the door to potential therapeutic interventions.

### 2.1. Dystroglycan

Dystroglycan was discovered in the embryonic chicken brain as the cell membrane glycoprotein able to bind laminin [13]. To date, it is considered a component of the dystrophin–glycoprotein complex (DGC), which links ECM proteins to the actin cytoskeleton via dystrophin in skeletal muscle cells and non-muscular tissues [14].

Dystroglycan is encoded by the *DAG1* gene and translated into a precursor protein (with a molecular mass of 97.5 kDa) that undergoes complex post-translational processing [15]. First, the precursor protein is cleaved at residue Ser-654 into two subunits named α-dystroglycan and β-dystroglycan. β-dystroglycan is located in the cell membrane; the N-terminal domain is extracellular, while the C-terminal domain is cytoplasmic. The N-terminal domain of β-dystroglycan binds the C-terminal of α-dystroglycan (completely located in the extracellular space) by non-covalent bonding. α-dystroglycan is a heavily glycosylated subunit and exhibits a mucin-type *O*-glycosylation site in the central region of the molecule. *O*-glycans of α-dystroglycan are crucial for laminin-binding [16]. Laminin requires calcium to establish additional coordination contacts with the sugar moieties protruding from the α-dystroglycan subunit [17]. Production of this laminin-binding glycan involves several glycosyltransferases, including POMT1, POMT2, POMGnT1, Fukutin, FKRP, and LARGE. The laminin-binding glycan on α-dystroglycan is designated as matriglycan, which is a glycosaminoglycan-like polysaccharide of [-3GlcAβ1,3Xylα1-] units [18]. Of note, engineering matriglycan alone can recapitulate dystroglycan binding and function [19].

The primary structure of dystroglycan is highly conserved in vertebrates and expressed in fetal and adult tissues, including adipose, epithelial, endothelial, and blood, even if it is mainly expressed by brain and skeletal muscle [20]. It belongs to a complex comprising several dystroglycan-associated proteins such as dystrophin, sarcoglycans (α, β, γ, and δ) syntrophins, and dystrobrevins (Figure 2).

The interaction of laminin with α-dystroglycan is considered crucial for the stability of BM. Most probably, tissue-specific glycosylation modifies the laminin-binding specificity of α-dystroglycan, as demonstrated by brain α-dystroglycan that presents unique glycoepitopes and preferential binding to laminin 10/11 [21].

Glycosylation defects can affect the ability of α-dystroglycan to bind laminin, leading to several human pathologies, in particular neuromuscular disorders. Specifically, hypoglycosylation of α-dystroglycan has been associated with a subgroup of rare genetic disorders named dystroglycanopathies. Dystroglycanopathies are characterized by muscular dystrophies, brain malformation, panoply of ocular defects, and intellectual disability [20]. In a mouse model of dystroglycanopathy, the administration of a bispecific antibody (biAb) that functions as a surrogate molecular linker to reconnect laminin-211 and β-dystroglycan improved muscle function [22]. Recently, the repeating hexasaccharide of the matriglycan was successfully and efficiently synthesized, opening the door for the reconstruction of muscle tissue in muscle disorders [23]. Deficiency of glycosylated α-dystroglycan was also associated with the depressive-like behaviors of male mice [24].

Upon binding to α-dystroglycan, laminin regulates cell shape by reorganizing the actin cytoskeleton, induces cell proliferation and differentiation, and modulates tissue-specific gene expression [20]. Since these activities are dysregulated during tumor progression, the loss function of dystroglycan could be the cause. Alterations in dystroglycan expression and glycosylation in human cancer cells have been well described elsewhere [20]. Interestingly, the loss of laminin binding to α-dystroglycan seems to play a crucial role in cancer pathogenesis. In prostate cancer, downregulation of LARGE2, a paralog of LARGE, resulted in hypoglycosylation of α-dystroglycan and loss of its ability to bind laminin 111, thus promoting cellular phenotypes associated with cancer progression and metastasis [25]. In rhabdomyosarcoma, α-dystroglycan lacks matriglycan modification and the ability to bind laminin. Ectopic expression of the glycosyltransferase LARGE1 restores matriglycan modifications and the ability of α-dystroglycan to bind laminin [26].

It is now clear that both dystroglycan expression and matriglycan synthesis on α-dystroglycan are reduced in many types of cancer and these changes could be associated with poor prognosis [20]. Since the alteration in the laminin-binding glycan of α-dystroglycan plays a crucial role in cancer development and progression, increasing expression of matriglycan could be a novel therapeutic approach for cancers, as demonstrated in breast cancer cells treated with ribitol-enhanced matriglycan [27].

In addition to its critical role in cell adhesion and cytoskeleton remodeling, α-dystroglycan serves as a main receptor for certain viruses. Pathogenic arenaviruses, like the Lassa virus (LASV), engage in high-affinity interactions with its specific glycan receptor, matriglycan [28]. Understanding the structural basis for matriglycan recognition by arenaviruses could lead to the design of new inhibitory molecules that compete for the receptor binding site and prevent viral entry [29]. For example, LARGE-dependent modification of dystroglycan at Thr-317/319 is required for laminin binding and arenavirus infection [30]. Moreover, Mycobacterium leprae invades Schwann cells by binding to the α-dystroglycan of Schwann cells via the interaction of α2-laminin in the basal lamina [31]. An in silico study was performed to select active compounds with the ability to block the α-dystroglycan and suppress LASV infection. Chrysin, reticuline, and 3-caffeoylshikimic acid were the top three ligands that formed the most stable interactions with α-dystroglycan and represent promising compounds to be further explored through in vitro and in vivo studies [32].

Recently, a neuroprotective effect has been assigned to laminin/dystroglycan interaction after hemorrhagic shock and reperfusion. This interaction modulates neuronal ferroptosis via the AMPK/Nrf2 pathway, offering a promising therapeutic strategy against the cognitive impairment induced [33].

### 2.2. 67-kDa Laminin Receptor

The 67-kDa laminin receptor (67 kDa LR) represents a multifunctional protein, ubiquitously expressed, located on the cell surface, in the cytosol and the nucleus. Traditionally, it is considered a major player in cell adhesion to BM by interacting directly with laminin. In addition to direct binding to laminin, 67 kDa LR facilitates interactions between laminin and integrins, acting as a co-receptor [34]. Beyond cell adhesion via laminin, 67 kDa LR exerts diverse functions, including ribosomal biogenesis and translation, pre-ribosomal RNA processing, cell migration, invasion, growth, cytoskeletal reorganization, and binding to histones and chromatin. These multiple activities are carried out through binding diverse molecular partners, including prion protein, and a variety of viruses and bacteria [35].

There is still no univocal nomenclature for 67 kDa LR. This mistake is caused by the complex structure of the receptor and the presence of the precursor protein. Indeed, this receptor was originally identified as a 67-kDa molecular weight peptide, whereas the precursor of this peptide, named 37-kDa laminin receptor precursor (37LRP), according to its molecular weight, was subsequently discovered. Thus, this receptor exists in two forms, 37LRP, which undergoes homo-dimerization and fatty acid acylation, and the resultant 67 kDa LR [35]. To date, the post-translational modifications responsible for the conversion of 37LRP into the higher molecular weight isoforms are not yet known. 37LRP is located primarily in the cytoplasm and nucleus, whereas 67 kDa LR is primarily in the lipid raft region of the plasma membrane [35,36]. Structurally, 67 kDa LR is composed of the intracellular N-terminal domain, a transmembrane domain, and an extracellular C-terminal domain. Three binding sites for the β1 chain of laminin have been identified: the region corresponding to 161–180 aa, also known as peptide G, the 209–229 aa sequence, and the most C-terminal domain containing the four acidic TEDWS-like repeats [37,38]. The below figure (Figure 3) shows the structure and the subunits involved in the interaction with the β1 chain of laminin.

Beyond its physiological role, 67 kDa LR is involved in tumor cell adhesion and migration to LM, crucial steps in tissue invasion and metastasis, by binding LM with high affinity. The inhibition of 67 kDa LR binding to laminin has been proven to be a promising approach to inhibit metastatic cancer cell spread. The 67 kDa LR-specific antibody IgG1-iS18 significantly reduces adhesion and invasion of metastatic lung, cervix, colon, and prostate cancer cells [39]. Moreover, the invasive potential of early- and late-stage colorectal cancer cells overexpressing 67 kDa LR was significantly blocked by IgG1-iS18 [40]. Recently, it has been revealed that the downregulation of 67 kDa LR by siRNA in lung cancer cells inhibits key cancer hallmarks, such as resistance to cell death, metastasis, and evasion of immune destruction [41]. In a prostate cancer cell line model, a novel compound against 67 kDa LR identified by an in silico approach exerts anti-tumor/anti-growth and anti-angiogenic effects [42].

In addition to tumor progression, 67 kDa LR plays a crucial role in the pathogenesis of Alzheimer’s disease by acting as a receptor for amyloid beta peptide (Aβ). Further, a link between 37/67 kDa LR and the processing of amyloid precursor protein (APP) has been found [43]. A significant reduction in Aβ levels and Aβ uptake was induced by the knockdown of 67 kDa LR with shRNA and IgG1iS18 specific antibody [44,45]. NSC48478, a small molecule targeting 37/67 kDa LR binding to laminin, was able to control APP maturation and intracellular localization in neuronal cells [46,47]. In human skin fibroblasts from familial Alzheimer’s disease, NSC48478 restores APP maturation and reduces Aβ levels [48]. In a genetic form of Alzheimer’s disease, the targeting of 37/67 kDa LR by NSC48478 inhibits the aberrant endocytosis of APP and improves homeostasis of the endocytic network and trafficking [49].

Interestingly, epigallocatechin-3-O-gallate (EGCG), a major component and the principal polyphenol in green tea, exerts beneficial effects in many diseases such as cancer, inflammatory diseases, and neurodegenerative disorders, by interacting with 67 kDa LR. Although the EGCG binding site to 67 kDa LR (161–170 aa) is separated by a short distance from the laminin-binding site (173–178 aa), EGCG inhibits the binding of laminin to 67 kDa LR [50,51]. It is probable that EGCG displays anti-tumoral, anti-inflammatory, and antioxidant effects as it prevents the binding of laminin to 67 kDa LR. Several clinical trials aimed to evaluate the beneficial effects of green tea are in progress or have been concluded. For example, in a phase 2 clinical trial, the clinical efficacy of the green tea extract polyphenon E has been demonstrated in patients with chronic lymphocytic leukemia [52]. It has been demonstrated that drinking three cups of green tea infusion per day produces beneficial effects on lipid profiles and HbA1c in patients with T2DM nephropathy [53]. Nevertheless, EGCG exerts beneficial effects via the 67 kDa LR; if the 67 kDa LR expression decreases, the effect of EGCG is inhibited. Since the expression level of 67 kDa LR may be different between individuals, further investigations are needed [54].

In contrast to such data, a beneficial effect resulting from 67 kDa LR/laminin interaction, elicited by treatment with YIGSR peptide, has been registered in lymphedema by promoting basement membrane repair and cell–cell adhesion [55].

Furthermore, 67 kDa LR does not only act as a receptor for laminin, but also as a receptor for viruses, such as Sindbis virus and Dengue virus, bacteria, cellular prion protein, and infectious prions [56]. It could be interesting to investigate the role of laminin in the formation of such molecular complexes on the cell surface. An analysis of the effects of inhibitory compounds targeting the binding of laminin to 67 kDa LR could be a feasible therapeutic approach for infectious diseases.

### 2.3. Syndecans

Syndecans are a small family of four transmembrane proteoglycans (syndecan-1 to -4), expressed on the surface of many cell types. Structurally, syndecans are composed of N-terminal signal peptide, an extracellular domain containing multiple consensus sequences for glycosaminoglycan (GAG) binding; protease cleavage sites allowing shedding; a single transmembrane domain; and C-terminal cytoplasmic domain [57]. The extracellular portion varies significantly between syndecan family isoforms, whereas the transmembrane and cytoplasmic domains are highly conserved [58]. Syndecan-1 is widely expressed in epithelial and plasma cells; syndecan-2 is expressed in mesenchymal cells, such as smooth muscle and fibroblasts; syndecan-3 is mainly expressed in the brain and nervous tissue, whereas syndecan-4 is broadly expressed [58].

Syndecans act as coreceptors of other cell surface receptors, such as integrins, growth factors, and chemokine receptors, thus leading to the initiation of receptor kinase activity and downstream signaling. At the same time, syndecans act as receptors for laminin. Several reports suggest that syndecans mediate the biological functions of laminin by direct binding to LG4-5 domains of laminin α chains [59]. Previously, it has been suggested that laminin uses syndecan receptors to promote cell attachment, whereas integrins for cell spreading [60], but this assumption has been questioned. The role of syndecans, both as receptors and coreceptors, is crucial for several cell functions such as cell proliferation, adhesion, migration, and angiogenesis. Beyond regulating the physiological function of laminin, syndecan/laminin interaction is involved in a variety of pathophysiological events, such as inflammation, wound healing, tumor growth, and angiogenesis [61]. Among the four syndecans, syndecans-1, -2, and -4 have been described as laminin receptors (Figure 4).

The interaction between syndecan-1 and laminin 332 is crucial for keratinocyte migration [62]. Recently, a novel mechanism underlying laminin 332-driven keratinocyte/ECM remodeling during wound repair has been found. Laminin 332 induces expression of the MMP-9 and MMP-14, which are crucial for their proteolytic activity within epithelial podosomes, by recruiting syndecan-1 [63].

The role of syndecans in tumor growth and metastasis has been established in a variety of cancer cell lines by using a synthetic peptide mimicking the active sequence, RKRLQVQLSIRT, from the LG4 domain of laminin-α1 chain, designated as AG73 [64]. In breast cancer cells, AG73 supports tumor cell adhesion and invasion through filopodia, by binding syndecans-1, -2, and -4 [65]. In addition, AG73 promotes invadopodia activity in human adenoid cystic carcinoma cells [66]. On the other hand, AG73 strongly promotes angiogenesis in the chick chorioallantois membrane (CAM) assay, as well as in tube formation and sprouting of aortic ring assay, thus resulting in a potent syndecan-binding angiogenesis stimulator [67]. For these reasons, AG73 could be useful for therapeutic application in ischemic injuries.

Given their altered expression and critical role in cancer progression, syndecans are attractive targets for cancer treatment, and several therapeutic approaches have been developed. Among them, BT062 (indatuximab ravtansine), targeting syndecan-1, shows promising clinical value in relapsed or refractory multiple myeloma, as demonstrated by phase 1/2a study [68]. Nimesulide, a non-steroidal anti-inflammatory drug selective for cyclooxygenase-2, causes cell cycle arrest in primary effusion lymphoma cell lines by syndecan-1 downregulation [69]. Interestingly, anoikis-resistant endothelial cells exhibit syndecan-4 overexpression accompanied by high invasive capacity and low adhesion to laminin. The downregulation of syndecan-4 promotes laminin adhesive capacity and reduces invasive capacity [70]. It has been found that trastuzumab, approved for the treatment of HER2-overexpressing breast and gastric cancer, and panitumumab, used for the treatment of patients with metastatic colorectal cancer, significantly decreased the expression of syndecan-4 [71,72]. This side effect could improve treatment efficacy.

Upon binding to syndecan receptors, laminins could be key drivers for the regulation of tissue morphogenesis. For example, laminin forms a macromolecular complex involving syntaxin-4 and syndecan-1 during mammary epithelial morphogenesis [73]. Syndecan-binding peptide-conjugated alginate hydrogel exerts beneficial effects in human nucleus pulposus cells from degenerated intervertebral discs [74]. Syndecan-4/laminin 332 interaction plays a crucial structural role also in human trabecular meshwork cells, as demonstrated by PEP75, a syndecan-4-binding peptide derived from laminin 332, which induces a cross-linked actin network necessary for controlling intraocular pressure [75].

### 2.4. Lutheran/Basal Cell Adhesion Molecule

Lutheran/basal cell adhesion molecule (Lu/BCAM) is a transmembrane protein belonging to the immunoglobulin superfamily of cell adhesion molecules. Originally identified in red blood cells, Lu/BCAM has been found in smooth muscle cells, endothelial cells, peripheral nerve cells, macrophages, and epithelial cells across various tissues [76]. Recently, it has also been detected in human hematopoietic stem cells [77].

On the cell surface, Lu/BCAM is expressed in two isoforms, named Lu (85 kDa) and BCAM (78 kDa), which differ in the size of the cytosolic domain. BCAM and Lu are two splice variants of the Lutheran antigen encoded by the Lu gene. Unlike Lu, BCAM is mainly expressed in the basal part of epithelial cells [76]. Structurally, Lu/BCAM is a single transmembrane glycoprotein composed of an extracellular domain with two variable-type and three constant-type domains; a transmembrane domain; and an intracellular domain (composed of 59 aa in Lu and 19 aa in BCAM) involved in binding to spectrin via RK573-574 motif and in cell signaling transduction [78]. On the plasma membrane, Lu/BCAM forms a molecular complex involving both laminin and spectrin [79]. Upon cleavage by membrane type-1-matrix metalloproteinase (MT1-MMP) at the juxtamembrane region of the extracellular domain, both Lu and BCAM can be released in a soluble form [80].

Lu/BCAM acts as a laminin receptor, thereby modulating different biological functions, including cell adhesion, migration, and invasion. Specifically, laminin containing the α5 chain (a constituent of laminin 511 and laminin 521) binds Lu/BCAM, through the D343 of domain 2–domain 3 interdomain region of the N-terminal immunoglobulin-like domains [81]. It is important to highlight that Lu/BCAM is the unique laminin receptor on the surface of circulating red blood cells in humans. The structure of the different isoforms of Lu/BCAM is shown in Figure 5.

The dysregulated adhesive function of Lu/BCAM has been found in many diseases, particularly in blood disorders. Particularly, Lu/BCAM mediates abnormal red blood cell adhesion to laminin in sickle cell disease and hereditary spherocytosis, due to the phosphorylation of the long isoform cytoplasmic domain or by its dissociation from the spectrin-based skeleton [81]. In polycythemia vera, red blood cells have abnormal expression of Lu/BCAM and hydroxycarbamide enhances Lu/BCAM phosphorylation and exacerbates cell adhesion to laminin [82]. Lu/BCAM expression is dysregulated in various types of cancer, such as skin, breast, lung, and bladder cancers, and could act as a biomarker and target for the treatment of these diseases [83,84,85]. Probably, Lu/BCAM promotes cell migration by competing with α3β1 integrin for binding laminin α5. This mechanism could affect integrin-mediated cell attachment to laminin α5 rather than promote cell anchoring [86]. In colorectal cancer, Lu/BCAM is overexpressed in clinical KRAS-mutant hepatic metastasis and the inhibition of its interaction with laminin α5 impaired adhesion of colorectal cancer cells to vascular endothelial cells, resulting in a reduction of metastatic growth [87]. In human bladder cancer cells, laminin stimulates cell adhesion through RhoA/Rac1 signaling pathway [85]. In this report, Lu/BCAM is considered oncogenic in human urothelial cancers and a promising novel therapeutic target.

No antibodies and/or small molecules targeting Lu/BCAM are currently under clinical investigation. However, an anti-human Lu/BCAM phage antibody has been recently identified. This antibody inhibits tumor cell migration on laminin-511 and it could be useful for the design of an antibody-drug to suppress cancer progression [88].

A pathogenetic role has also been assigned to soluble Lutherans. In fact, in vitro studies demonstrated that soluble Lutheran isolated from hepatocellular carcinoma cells can bind laminin 511 and modulate cell migration on laminin 511 [80]. In addition, Lu/BCAM may act as a tumor suppressor gene, as demonstrated in thyroid cancer where Lu/BCAM is downregulated and negatively correlated with tumor growth [89].

Recently, it has been reported a new mechanism by which Lu/BCAM binding to laminin α5 is regulated in erythrocytes. Specifically, Lu/BCAM is gradually activated during erythrocyte aging due to the loss of sialic acid on glycophorin-C. Upon activation, it engages a sialic acid-dependent interaction with laminin-α5 [90]. These data shed new light on the mechanisms contributing to the increased adhesiveness of senescent erythrocytes, probably facilitating their clearance.

### 2.5. Melanoma Cell Adhesion Molecule

Melanoma cell adhesion molecule (MCAM), also known as CD146 or cell surface glycoprotein MUC18, is a component of the Ig superfamily expressed in several cell lines, such as vascular endothelial cells, smooth muscle cells, glomerular mesangial cells, Schwann cells, mesenchymal stem cells, and leukocytes [91]. It was originally identified as a tumor marker for melanoma, but subsequent evidence has demonstrated that its expression is related to tumor progression and poor survival also in prostate cancer, osteosarcoma, hepatocellular carcinoma, and other tumors [92].

There are three forms of MCAM proteins in humans: the short and the long membrane-anchored forms, encoded by the cd146 gene and generated by alternative splicing of the transcript in exon 15; and the soluble form of MCAM that is produced by the proteolytic cleavage of the membrane forms [91]. The long membrane isoform is primarily located at the cell junction and is involved in structural functions, whereas the short isoform is expressed at the apical membrane of the cell and is involved in angiogenesis [93,94]. Even if the crystal structure is not yet available, MCAM is an integral membrane glycoprotein composed of an extracellular portion with five distinct Ig-like domains (V-V-C2-C2-C2 structural motif), a hydrophobic transmembrane section, and a cytoplasmic tail that differs between the two membrane-anchored forms [95,96]. MCAM is mainly a monomeric protein that dimerizes in response to physiological stimuli [97]. Figure 6 shows the structure of the different isoforms of MCAM and the subunits involved in the interaction with the β1 chain of laminin.

MCAM could not be considered solely as an adhesion receptor but is a cell surface receptor for numerous ligands. In fact, MCAM ligands have been categorized into three groups: components of ECM, pro-angiogenic receptors, and growth factors [91]. Here, we discuss the implications of MCAM/laminins interaction in the physio-pathological context.

The main interactors of MCAM are α4-laminins, such as laminins 411 and 421, which are expressed by mesenchymal cells and promote adhesion and migration of the cells. Under pathological conditions, α4-laminins are expressed by tumor cells and strongly promote tumor cell migration [98].

Flanagan et al. discovered an important mechanism through which MCAM promotes pathogenetic autoimmune response in multiple sclerosis. CD4^+^ T helper cells express MCAM and use the binding to laminin 411 to enter the central nervous system (CNS). In an animal model of multiple sclerosis, an antibody targeting MCAM reduced Th17 cell infiltration into the CNS, thus ameliorating experimental autoimmune disease [99]. Further, MCAM-laminin 411 interaction facilitates trans-endothelial migration of MCAM^+^ T cells into the CNS over the choroid plexus [100]. Recently, Charabati et al. has concluded that MCAM is the pathological pathway used by brain endothelial cells to recruit pathogenic CD4^+^ T lymphocytes from circulation during early neuroinflammation [101]. In psoriatic arthritis, MCAM^+^ T cells are enriched at synovial inflammation sites and are critical for the disease process [102].

While MCAM^+^ lymphocytes bind to laminin 411 to enter tissues and promote inflammatory and autoimmune reactions, MCAM^+^ cancer cells, including melanoma cells, bind to laminin 421 to induce cancer metastasis via vascular and lymphatic vessels [103]. In tumor blood vessels of renal cell carcinoma, MCAM and LAMA4 are the most consistently expressed genes and predict patient outcomes [104].

Given that higher expression level of MCAM on human osteosarcoma cells than HER2 and EGFRA, a radiolabeled monoclonal antibody targeting MCAM (^177^Lu-labeled OI-3) has been evaluated to target the osteosarcoma metastases and circulating tumor cells. The results have shown that ^177^Lu-labeled OI-3 is a valid candidate for radioimmunotherapy against MCAM^+^ cancers [105]. Another study has highlighted the potential of MCAM-targeted radioimmunotherapy in malignant mesothelioma by using CD146-targeting antibody OI-3 coupled with ^212^Pb (^212^Pb-TCMC-OI-3) [106].

## 3. Discussion

Laminins are one of the most important glycoproteins of BM, composed of three disulfide-linked polypeptides, the α, β, and γ chains. Specific chain composition led to the assembly of 16 laminin isoforms. The laminin composition and architecture vary in a tissue- and cell-specific fashion. In addition, the expression of laminin isoforms is differently regulated during development and pathological conditions. At the end of the long arm, the C-terminus of the α chain folds into five globular domains, LG1 to LG5, which are crucial for interacting with multiple receptors in the plasma membrane of cells and microorganisms. Upon interaction with specific receptors, laminin critically contributes to cell attachment and differentiation, cell migration, and cell survival.

Laminin receptors involve a variety of cell surface receptors that could be categorized into two classes, integrin and non-integrin receptors. The binding site of laminin for integrins is located in LG1-3 domains whereas LG4-5 domains represent the binding site of laminin for non-integrin receptors. The expression and functions of integrin receptors for laminin have been extensively investigated, while the functional involvement of non-integrin receptors in health and disease is still unresolved. Of course, cooperation between integrin and non-integrin receptors in the regulation of laminin functions is conceivable, but not yet confirmed.

The present review recapitulates for the first time the main structural and functional aspects of non-integrin receptors for laminins including dystroglycan, 67-kDa laminin receptor (67 kDa LR), syndecans, Lutheran/basal cell adhesion molecule (Lu/BCAM), and Melanoma cell adhesion molecule (MCAM), under both normal and diseased conditions.

Dystroglycan is an ECM receptor undergoing extensive posttranslational processing and O-glycosylation that is required for laminin binding and this binding is mediated by a glycan structure known as matriglycan. Dystroglycan/laminin interaction is essential for skeletal stability during muscle contraction, providing a “molecular bridge” between cells and the surrounding tissues. In fact, disruption of this interplay underlies a group of inherited muscular dystrophies. Moreover, reduction or loss of matriglycan is involved in cancer initiation and progression and is significantly associated with poor prognosis. Therefore, enhancing or stimulating the expression of matriglycan could be considered potential therapeutics.

While 67 kDa LR is classically considered a cell surface receptor for laminin, it also acts as a receptor for elastin, carbohydrates, microorganisms, and prion proteins. Furthermore, 67 kDa LR is widely investigated in the context of cancer where it is overexpressed and significantly associated with the invasiveness and metastatic potential of tumor cells. An interesting field of research concerns the role of 67 kDa LR in neurodegenerative disorders, such as Alzheimer’s disease. Beyond its cell-adhesive properties resulting from binding to laminin, 67 kDa LR is involved in amyloid precursor protein (APP) metabolism and β-Amyloid internalization that represents a key aspect of Alzheimer’s disease progression. In fact, internalized β-Amyloid induces cell dysfunction and causes neuronal death. These assumptions suggest 67 kDa LR as a potential target for therapeutic intervention in neurodegenerative disorders. Interestingly, the activation of 67 kDa LR leads to different signaling pathways depending on the binding of a specific agonist. For example, 67 kDa LR/laminin interaction drives toward proliferative signaling, resistance to growth suppressors and cell death, angiogenesis, migration, and metastasis. On the other hand, green tea polyphenol EGCG performs antioxidant, anti-inflammatory, and anti-tumoral effects by interacting with 67 kDa LR and displacing laminin-binding. Overall, these data demonstrate that 67 kDa LR is a versatile receptor, and the phenomenon of “biased agonism” is conceivable. This hypothesis could be a starting point for future studies.

Syndecans 1, 2, and 3 are syndecan isoforms that act as cell surface receptors for laminins. In addition to being the central mediator of cell adhesion, syndecans play pivotal roles in the initiation and progression of many diseases, particularly cancer. The clustering of syndecans with laminin triggers signaling cascades and regulates several cell functions in the tumor microenvironment and in the context of tumor angiogenesis.

Lu/BCAM is the unique laminin receptor expressed on the red blood cell surface. In these cell types, dysregulation of Lu/BCAM contributes significantly to the pathogenesis of hematological disorders by influencing the adhesion between cells and laminin. In tumor cells, Lu/BCAM enhanced tumor cell migration ability by modulating integrin-mediated cell attachment to laminin. It is conceivable that Lu/BCAM is involved differently in solid and liquid tumors or hematological disorders. In solid tumors, Lu/BCAM could reduce cell adhesion to laminin and increase cell motility. Other signaling pathways could induce phosphorylation modification of Lu/BCAM, thereby affecting its adhesion to laminin. Competition mechanism between Lu/BCAM and integrins has also been hypothesized.

MCAM represents a valid tumor marker in several types of cancer. Interestingly, MCAM is an emerging factor affecting neuroinflammation. MCAM^+^ brain endothelial cells are responsible for the recruitment of CD4^+^ T lymphocytes from circulation. The targeting of this mechanism is a promising therapeutic approach for multiple inflammatory disorders.

Collectively, this review provides a detailed examination of non-integrin receptors, crucial components of the complex mechanism that governs cell interactions with ECM proteins. Upon binding to laminin, non-integrin receptors can signal independently or may associate with integrins and growth factor receptors to induce cell signals, thus affecting cell proliferation, differentiation, adhesion, and migration.

Since non-integrin receptors are characterized by a wide structural and biological diversity, tissue specificity, functional outputs, molecular pathophysiology, and the strategies to therapeutically target these receptors are summarized in Table 1.

## Figures and Tables

**Figure 1 ijms-26-03546-f001:**
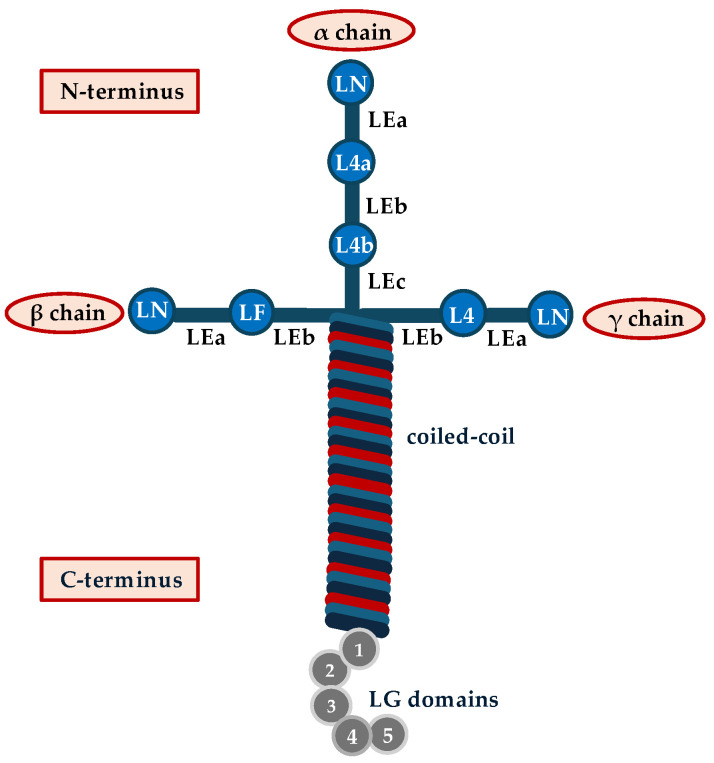
**Form and structure of laminin**. The domain organization of laminin-111 is used as an example of laminin structure. At the N-terminus, the α1 chain contains an LN domain followed by LE, L4, and a coiled-coil domain. A similar structural organization characterizes the N-terminus of β and γ subunits. At the C-terminus, the α1 chain contains five globular (LG) domains that are absent in β and γ subunits. The organization of these chains determines a peculiar cruciform or cross structure characterized by one long and three short arms.

**Figure 2 ijms-26-03546-f002:**
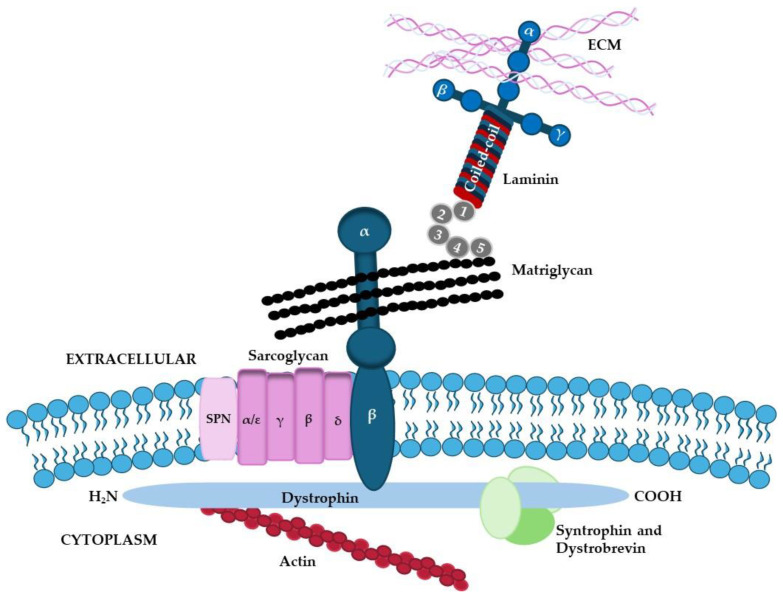
**Schematic illustration of the structure of dystroglycan and the molecules that bind to its α and β chains.** β-dystroglycan is located in the cell membrane whereas α-dystroglycan is completely located in the extracellular space. α-dystroglycan is a heavily glycosylated subunit and exhibits a mucin-type O-glycosylation site in the central region of the molecule. O-glycans of α-dystroglycan are crucial for laminin-binding. The laminin-binding glycan on α-dystroglycan is designated as matriglycan.

**Figure 3 ijms-26-03546-f003:**
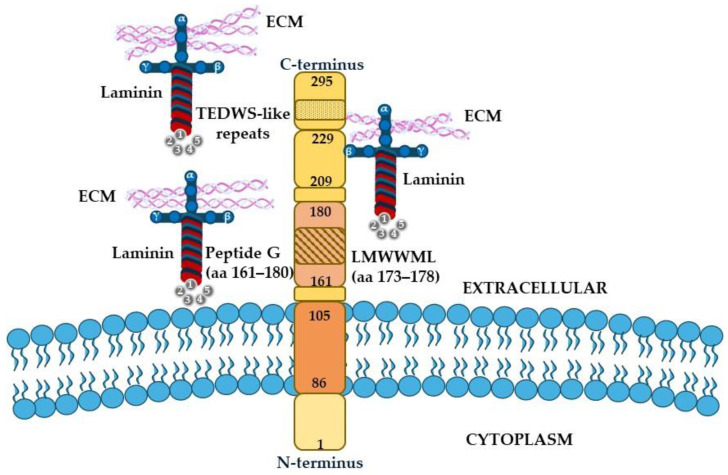
**Schematic representation of the 67-kDa laminin receptor.** 67-kDa laminin receptor is a type II transmembrane receptor of 295 aa located in the lipid raft region of the plasma membrane. The receptor is divided into three domains, the intracellular N-terminal domain, transmembrane domain, and extracellular C-terminal domain. The three binding sites for the β1 chain of laminin are the region corresponding to 161–180 aa, also known as peptide G, the 209–229 aa sequence, and the most C-terminal domain containing the four acidic TEDWS-like repeats.

**Figure 4 ijms-26-03546-f004:**
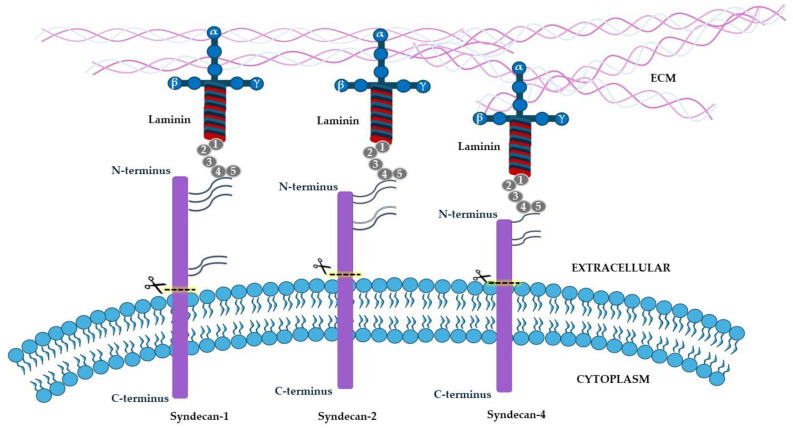
**Schematic representation of syndecan-1, -2, and -3 as laminin receptors**. Syndecans are four transmembrane proteoglycans composed of N-terminal signal peptide, an extracellular domain containing multiple consensus sequences for glycosaminoglycan (GAG) binding; protease cleavage sites allowing shedding; a single transmembrane domain; and C-terminal cytoplasmic domain.

**Figure 5 ijms-26-03546-f005:**
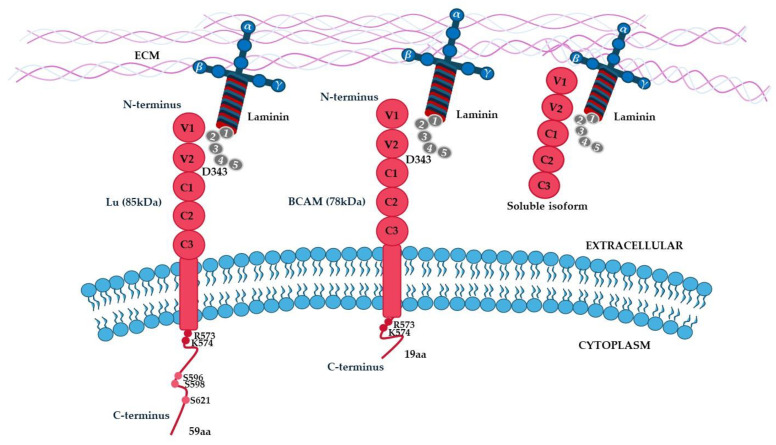
**Schematic representation of (Lu/BCAM).** On the cell surface, Lu/BCAM is expressed in two isoforms, named Lu (85 kDa) and BCAM (78 kDa), which differ in the size of the cytosolic domain. Lu/BCAM is a single transmembrane glycoprotein composed of an extracellular domain with two variable-type and three constant-type domains, a transmembrane domain, and an intracellular domain (composed of 59 aa in Lu and 19 aa in BCAM) involved in binding to spectrin via RK573-574 motif and in cell signaling transduction. Upon cleavage by membrane type-1-matrix metalloproteinase (MT1-MMP) at the juxtamembrane region of the extracellular domain, both Lu and BCAM can be released in a soluble form.

**Figure 6 ijms-26-03546-f006:**
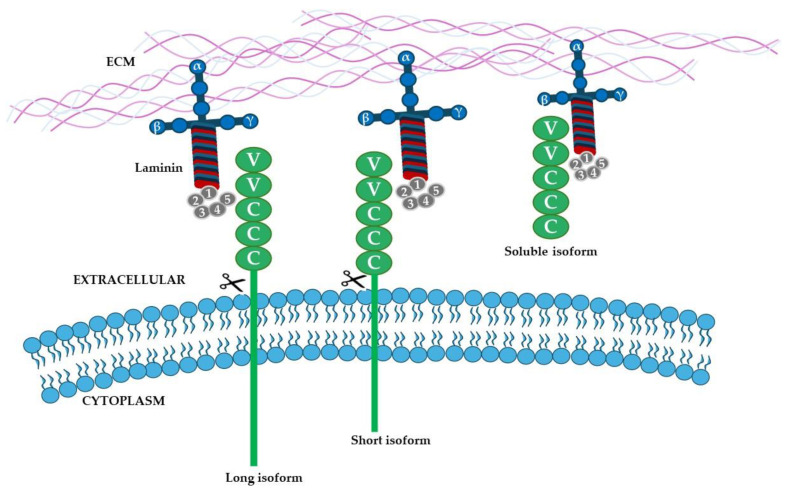
**Schematic representation of MCAM.** There are three forms of MCAM proteins: the short and the long membrane-anchored forms, encoded by the cd146 gene and generated by alternative splicing of the transcript in exon 15; and the soluble form of MCAM that is produced by the proteolytic cleavage of the membrane forms.

**Table 1 ijms-26-03546-t001:** Summary of the main features of non-integrin receptors.

Non-Integrin Receptors	Tissue Specificity	Functional Outputs	Related Diseases	Active Compounds or Neutralizing Antibodies
**Dystroglycan**	adipose, epithelial, endothelial, blood, brain and skeletal muscle	stability of basement membrane, cell shape, proliferation, differentiation, tissue-specific gene expression	Dystroglycanopathies Cancer Arenavirus infections	Ribitol-enhanced matriglycan
**67-kDa LR ***	ubiquitous	cell adhesion, ribosomal biogenesis and translation, pre-ribosomal RNA processing, cell migration, growth, cytoskeletal reorganization	Cancer Alzheimer’s disease Microbial and viral diseases	IgG1-Is18 NSC48478 Epigallocatechin-3-O-gallate YIGSR peptide
**Syndecan-1**	epithelial and plasma cells	cell proliferation, adhesion, migration, wound healing, mammary epithelial morphogenesis	Cancer (multiple myeloma)	AG73 peptide BT062 (indatuximab ravtansine)
**Syndecan-2**	mesenchymal cells, fibroblasts, and smooth cells	cell proliferation, adhesion, and migration	Cancer	AG73 peptide
**Syndecan-4**	ubiquitous	cell proliferation, adhesion, and migration, control of intraocular pressure	Cancer	AG73 peptide PEP75
**Lu/BCAM ****	red blood, smooth muscle, endothelial, peripheral nerve, epithelial cells, macrophages, and hematopoietic stem cells	cell adhesion, migration, and invasion	Sick cell disease Hereditary Spherocytosis Polycythemia vera Cancer	-
**MCAM *****	vascular endothelial, smooth muscle, glomerular mesangial, Schwann, mesenchymal cells, and leukocytes	cell adhesion and migration	Cancer Multiple sclerosis Psoriatic arthritis	^177^Lu-labeled OI-3 ^212^Pb-TCMC-OI-3

* 67-kDa laminin receptor. ** Lutheran/basal cell adhesion molecule. *** Melanoma cell adhesion molecule.

## Data Availability

Not applicable.

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
