# Peer review of "Non-Integrin Laminin Receptors: Shedding New Light and Clarity on Their Involvement in Human Diseases"

_ijms, 2025, doi:10.3390/ijms26083546_

Round 1

Reviewer 1 Report

Comments and Suggestions for Authors

The manuscript presents a comprehensive review of non-integrin receptors for laminins, emphasizing their structural characteristics and functional roles in various physiological and pathological contexts. However, several aspects could be improved to enhance clarity and readability for the reader:

1- The structure of laminin as itself should be clearly and separately described for better understanding.

2- Instead of consolidating all non-integrin laminin receptors into a single figure, the manuscript could present each receptor’s structure along with its associated signaling pathways dedicated under their subheadings. This approach would improve organization and clarity.

3- Some sections use inconsistent terminology (e.g., "matryglycan" vs. "matriglycan"). Standardizing the terminology throughout the manuscript would improve coherence and readability.

Reviewer 2 Report

Comments and Suggestions for Authors

The manuscript titled “Non-integrin laminin receptors: shedding new light and clarity on their involvement in human diseases” summarized recent findings about non-integrin laminin receptors and their roles in human diseases. Laminin is a crucial component in extracellular matrix, and have important functions as structure molecule and signaling molecule. However, more research attention is focused on collagens and their receptors, compared to laminin. Therefore, it is necessary for the authors to review the importance of laminin receptors, and to highlight their involvement in human diseases. Overall, this review is well written. Before publication, the authors should consider further improvement at the following points as I listed here.

  1. There are numerous examples how researches of integrins are translated to pharmaceutical compounds for human diseases. Readers might be interested in whether non-integrin laminin receptors have similar potential as drug targets. The authors should add more information about this topic, such as evaluation results of pre-clinical models for active compounds or neutralizing antibodies targeting these receptors.
  2. It is interesting that diverse types of receptors recognize laminin, but they trigger different downstream signaling pathways. The authors introduced detailed information of these receptors one by one as separate chapters. A chart should be used to inspire comprehensive understanding and comparison of these receptors regarding their structures, tissue specificity, functions, related diseases and so on.
  3. The dysregulation of laminin, instead of their receptors, is also involved in many types of human diseases, and the interaction between laminin and receptors could also be pharmaceutical targets in such conditions. The authors should expand the topic and include these diseases in this review.
